# Rashba-splitting-induced topological flat band detected by anomalous resistance oscillations beyond the quantum limit in ZrTe$_5$

Dong Xing[1,2], Bingbing Tong[1], Senyang Pan [3], Zezhi Wang[1,2], Jianlin Luo[1,2], Jinglei Zhang[3] & Cheng-Long Zhang [1] ✉

Topological flat bands − where the kinetic energy of electrons is quenched − provide a platform for investigating the topological properties of correlated systems. Here, we report the observation of a topological flat band formed by polar-distortion-assisted Rashba splitting in the three-dimensional Dirac material ZrTe$_5$. The polar distortion and resulting Rashba splitting on the band are directly detected by torque magnetometry and the anomalous Hall effect, respectively. The local symmetry breaking further flattens the band, on which we observe resistance oscillations beyond the quantum limit. These oscillations follow the temperature dependence of the Lifshitz−Kosevich formula but are evenly distributed in B instead of 1/B at high magnetic fields. Furthermore, the cyclotron mass gets anomalously enhanced about $10^2$ times at fields ~ 20 T. Our results provide an intrinsic platform without invoking moiré or order-stacking engineering, which opens the door for studying topologically correlated phenomena beyond two dimensions.

Flat electronic bands harbor exotic quantum behaviors due to the quenched kinetic energy and subsequently dominated Coulomb interaction. The fractional quantum Hall effect[1,2] is an archetypical two-dimensional (2D) flat band system. Recently developed moiré-engineered 2D twisted bilayer graphene[3–5] and multilayer graphene in certain stacking order[6–8] are other examples of realizing topological flat bands. Flat bands are also theoretically predicted in some stoichiometric three-dimensional (3D) materials forced by certain geometric lattices, like the kagome or Lieb lattices[9,10]. Flat band-induced correlation in 3D topological systems is crucial for realizing correlated 3D topological effects, like correlation on Weyl semimetals (WSM) and possible axionic dynamics[11–13]. Despite theoretical advancement in establishing the material database of topological flat bands[14], experimental realization of an isolated topological flat band around Fermi level (FL) in 3D stoichiometric materials remains elusive.

In the twisted bilayer graphene, the two preconditions for realizing a 2D topological flat band are: 1. the pristine Dirac material graphene; 2. moiré superlattice as the method for flattening the band. We now ask a question: can we find a counterpart in 3D? If there is, then which material is the 3D counterpart of graphene? How do we flatten the energy band or enlarge the unit cell in 3D? Here, we report that ZrTe$_5$[15], as a typical Dirac material in 3D, can meet the first precondition; the polar distortion at low temperatures in ZrTe$_5$ meets the second precondition without invoking van der Waals heterostructure-based engineering. The polar distortion-assisted Rashba splitting in ZrTe$_5$ is evidenced by torque magnetometry and the anomalous Hall effect (AHE), which shows the existence of a topological flat band in the magnetic field, on which anomalous resistance oscillations appear beyond the quantum limit. The cyclotron mass is enhanced by an order of $10^2$, consistent with the topological flat band. Our work also

[1]Beijing National Laboratory for Condensed Matter Physics, Institute of Physics, Chinese Academy of Sciences, Beijing 100190, China. [2]School of Physical Sciences, University of Chinese Academy of Sciences, Beijing 100049, China. [3]High Magnetic Field Laboratory, HFIPS, Chinese Academy of Sciences, Hefei 230031, China. ✉e-mail: chenglong.zhang@iphy.ac.cn

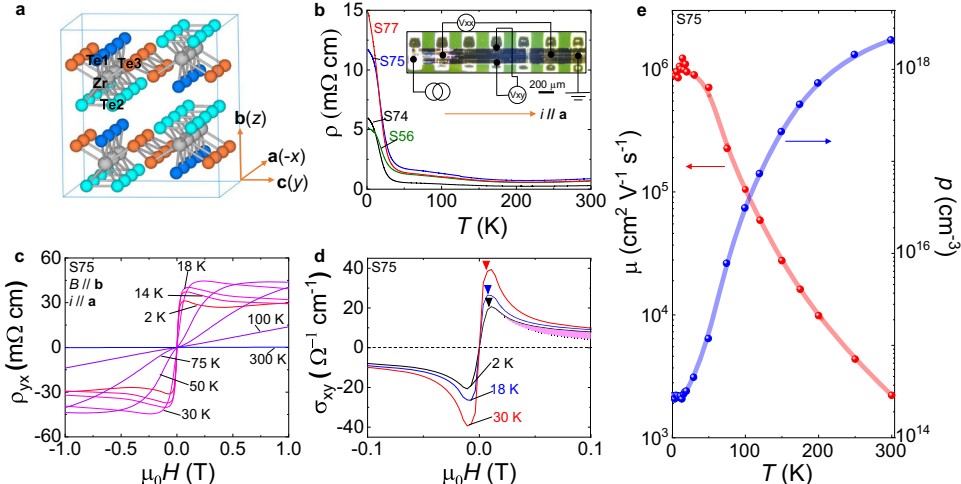

**Fig. 1 | Electrical transport characterizations of flux-grown ZrTe$_5$. a** Crystal structure of undistorted ZrTe$_5$. **b** Temperature-dependent resistivity $\rho_{xx}$ of typical flux-grown ZrTe$_5$ samples measured in this work. The inset shows an image of a typical device with patterned Au electrodes. **c** Temperature-dependent Hall resistivity $\rho_{yx}$ of sample S75. Anomalous term develops when the temperature is low.

**d** Temperature-dependent Hall conductivity $\sigma_{xy}$ converted from $\rho_{yx}$. The anomalous term $\sigma_{xy}^A$, shadowed by the pink area, occurs after the Drude subtraction (dashed line). **e** Temperature-dependent carrier density ($p$) and mobility ($\mu$) of sample S75.

highlights the importance of local symmetry breaking in topological materials.

## Results

### Rashba splittings and anomalous Hall effect in ZrTe$_5$

ZrTe$_5$ was initially proposed as a typical candidate for a quantum spin Hall insulator in 2D limit[15]. As shown in Fig. 1a, 2D ZrTe$_5$ layers stack along **b** axis (here, we use **a**, **b**, **c** and -x, z, y interchangeably) with a ZrTe$_3$ chain that runs along **a** axis. The topological property of 3D ZrTe$_5$ is sensitively dependent on the inter- and intra-layer coupling strengths. ZrTe$_5$ is located near the boundary of weak topological (WTI) and strong topological insulators (STI) as typical Dirac material[15], promoting many exotic phenomena[16–22]. Therefore, the properties of ZrTe$_5$ are sensitively dependent on crystal growth methods, namely the chemical vapor transfer (CVT) and flux processes. Samples grown from Te-flux are more stoichiometric and closer to the phase boundary[23–26]. As shown in Fig. 1b, temperature-dependent resistivity measured on flux-grown single crystals (see Supplementary Fig. 1 for more detailed characterizations of single crystals) exhibits semiconducting-like behavior, a typical profile in narrow-gapped semiconductors[27,28]. We focus on electrical transport within **ac** plane with current **i** along **a** axis as illustrated in the inset of Fig. 1b. Hall measurements (Fig. 1c) show that hole is the only carrier down to 2 K. Hall conductivity $\sigma_{xy}$ (Fig. 1d) fitted by Drude model $\sigma_{xy}^{Drude} = \frac{pe\mu^2 B}{1+\mu^2 B^2}$, where $p$ is the carrier density and $\mu$ is the mobility, shows the existence of an anomalous term $\sigma_{xy}^A$, as indicated by the pink shadowed area (see Supplementary Fig. 2 for more details on the Drude fittings). As shown in Fig. 1e, $p$ is ultralow ~ $3 \times 10^{14}$ cm$^{-3}$ and $\mu$ is as high as $10^6$ cm$^2$ V$^{-1}$ s$^{-1}$ at low temperatures, crucial for realizing the topological flat band and anomalous resistance oscillations. Therefore, ZrTe$_5$ sample synthesized in this work is a 3D counterpart of graphene.

We then investigate the possible local modifications on the Dirac band, which might provide the clue for forming a topological flat band. In these flux-grown samples, preliminary evidence of polar distortion is reported by nonlinear transport[26], while direct evidence is still lacking. We adopted torque magnetometry to measure the magnetic susceptibility tensor $\chi_{ij}$ defined by $\mathbf{M}_i = \chi_{ij}\mathbf{H}_j$, where $\mathbf{M}_i$ is the magnetization. $\chi_{ij}$ directly reflects the underlying point group symmetries due to Neumann's principle. Magnetic torque is defined as $\boldsymbol{\tau} = \mu_0 V \mathbf{M} \times \mathbf{H}$, where $\mu_0$ is the vacuum permeability, and $V$ is the volume of the sample. The space group of ZrTe$_5$ is $Cmcm$ (No. 63) with

a point group $D_{2h}$, under which the only permitted tensor elements in $\chi_{ij}$ are $\chi_{aa}, \chi_{cc}$ and $\chi_{bb}$. In our torque setup (inset of Fig. 2a), the cantilever picks up $\boldsymbol{\tau}_{a(x)} = \boldsymbol{\tau}_{2\theta} = A_1 \sin 2\theta$, where $A_1 = \frac{1}{2}\mu_0 V H^2 (\chi_{cc} - \chi_{bb})$. Therefore, we anticipate a pure $\sin 2\theta$ relation in angle-dependent $\boldsymbol{\tau}_a$. As shown in Fig. 2a, $\boldsymbol{\tau}_a$ at 7 T globally shows $\pi$ periodicity consistent with $\sin 2\theta$ relation. However, the negative and positive amplitudes, noted as Amp+ and Amp− in Fig. 2a, show asymmetry against a pure $\sin 2\theta$ relation, which is absent in a CVT sample (see Supplementary Fig. 3b and Supplementary Note 1). Further, we find the torque signal at 4 K can be well fitted by $\boldsymbol{\tau}_a = \boldsymbol{\tau}_{2\theta} + A_2 \sin^2 \theta$. The appearance of $A_2$ term directly shows the original orthorhombic symmetry is broken at low temperature, which is consistent with polarity (**P**) along the out-of-plane **b** axis (**P**//**b**) in our nonlinear transport results (see Supplementary Fig. 4 and Supplementary Note 2 for details of symmetry analyses). We now show that the AHE is the direct consequence of this polar distortion. As shown in Fig. 2b, $\sigma_{xy}^A$ starts to show up and exhibits plateau-like structures beyond a critical magnetic field B$_p$. As shown in Fig. 2c, we extract the temperature-dependent coefficients $A_1, A_2$, and $\sigma_{xy}^A$, and plot the ratios of $A_{1,2}(T)/A_{1,2}(270K)$ and $\sigma_{xy}^A$ together. $A_1(T)/A_1(270K)$ exhibits moderate temperature dependence and dominates over the total torque $\tau_a$ in the whole temperature range (see Supplementary Fig. 3a for the raw data). However, $A_2(T)$ is almost negligible at temperatures higher than 150 K, below which the ratio of $A_2(T)/A_2(270K)$ suddenly gets enhanced and reaches a value of ~30 at 2 K, indicating the emergence of polar distortion at 150 K. Furthermore, as shown in Fig. 2c, $\sigma_{xy}^A$ shows up around 150 K, where the $A_2$ term suddenly gets enhanced. The concurrence of the polar distortion and $\sigma_{xy}^A$ indicates that the AHE is locked to the polar distortion-induced band modifications, which is also consistent with the fact that no AHE is observed in CVT samples[20].

ZrTe$_5$ is a nonmagnetic material with time-reversal symmetry. Nevertheless, the observed AHE takes a profile like the AHE of ferromagnets with saturating plateaus. Based on our three observations: **1**. The existence of polar distortion **P**//**b**; **2**. The coincidence of the onsets of the AHE and polar distortions; **3**. The absence of the in-plane Hall effect in our samples (see Supplementary Fig. 5 and Supplementary Note 3 for the in-plane Hall discussions). Considering the quasi-2D nature of ZrTe$_5$, we interpret this behavior by invoking a Rashba model-based mechanism usually adopted for explaining the intrinsic AHE in magnetic materials[29]. With polar distortion **P** along **b** axis, the typical Rashba model appears as $H = \boldsymbol{\alpha} \cdot (\boldsymbol{\sigma} \times \mathbf{k}) + \Delta(B)\sigma^z$ with the time-

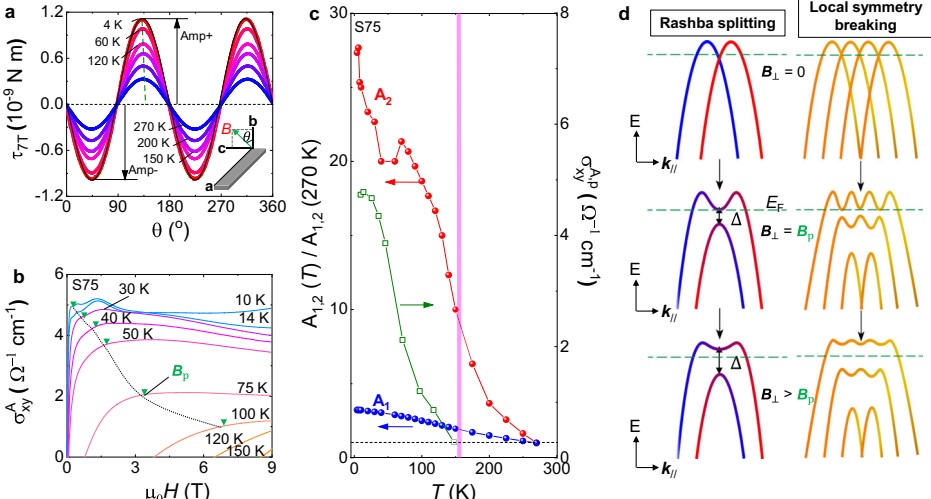

**Fig. 2 | Polar distortion-induced Rashba band splittings in ZrTe₅. a** Angle-dependent magnetic torque at 7 T measured at different temperatures. The inset illustrates the experimental setup. The solid black line shows the fitting by the formula $\tau_a = A_1 \sin 2\theta + A_2 \sin^2\theta$, where $A_1$ represents the orthorhombic structure and $A_2$ represents the lower-symmetric structure. **b** Temperature-dependent anomalous Hall conductivity $\sigma_{xy}^A$ obtained after subtracting the Drude component. $B_p$ is defined as a critical magnetic field of the onset of plateau-like structure on $\sigma_{xy}^A$.

**c** Temperature-dependent $A_{1,2}(T)/A_{1,2}(270K)$ and $\sigma_{xy}^A$ in sample S75. $A_{1,2}(T)$ denotes $A_1$ and $A_2$. The dashed vertical pink line indicates the concurrence of $\sigma_{xy}^A$ and polar distortion. **d** Illustration of band modifications in the presence of polar distortion. The left-hand column shows the magnetic field-induced gap in the original Rashba bands, and finally, a local topological flat band is formed at the top of the band in a higher magnetic field. The right-hand column shows the flatness is strongly enhanced by the local symmetry breaking promoted by polar distortions.

reversal breaking term $\Delta(B)$, here momentum **k** in the **ac** plane, $\alpha$ is the strength of Rashba splitting, and $\sigma$ is the Pauli matrix for real spins. Furthermore, we do not include $k_z$ dispersion in the Rashba model due to the parabolic relation along $k_z$[30]. As shown in Fig. 2d, the band splitting is very weak, and the crossing point is near the band edges due to tiny polar distortions, which means only samples with ultralow carrier density can access this region. When an out-of-plane magnetic field ($B_\perp$) is applied, the crossing point on the Rashba bands is gapped out due to a significant Zeeman effect $\Delta(B)$, the region near the gap becomes Berry curvature hot spots. With increasing magnetic field, the FL located near the band edge falls into the Rashba gap at $B_p$, results in a saturated $\sigma_{xy}^A$ in higher fields[29]. When we tilt the direction of a magnetic field to the **ac** plane, no in-plane Hall signal is detected (Supplementary Fig. 5a, b), which is consistent with the Rashba model, because the in-plane magnetic field can only shift the Rashba crossing without opening a gap, then no AHE is expected to appear.

The prominent feature of the formed Rashba band is that the extremum of the gapped band gets flatter with increasing magnetic field, creating an ideal, isolated topological flat band around FL, different from the proposal of NLSM-based flat band used to explain the results of nonlinear transport[31]. The origin of tiny polar distortions in the flux-grown ZrTe₅ sample is not yet clear because no structural transition is observed by x-ray diffraction down to 10 K[26], which might also indicate this weak polarity **P** is unable to drive the parent TIs to WSM phase via the Murakami's scheme[32]. As we know, the possible defects and disorder-induced polar distortion, as essential roles of local symmetry breakings, will lift the valley or spin degeneracies, and form a supercell[33–35]. As the main result of this work, we propose in this work that the effect of polar distortion together with Rashba splitting on the band edge, by forming the supercell, is similar to moiré engineering. As shown in the right column of Fig. 2d, the local symmetry breaking enlarges the unit cell, induces multiple Rashba splittings, and creates a topological flat band.

## Anomalous resistance oscillations beyond the quantum limit

As we will show, the flatness on the modified band is further supported by the observation of anomalous resistance oscillations beyond the quantum limit. The carrier density of sample S75 is $3 \times 10^{14}$ cm⁻³ at 2 K, corresponding to a quantum limit less than 0.05 T (**B** ∥ **b** axis), beyond

which there are no quantum oscillations. However, as shown in Fig. 3a, b, we observe strong resistance oscillations in sample S75, and reproduced in another sample S74, where similar behaviors persist up to 18 T (Fig. 3c). As noted by blue vertical lines in Fig. 3c, we find evenly distributed oscillations at fields higher than 3 T, which is clearly exhibited in $\Delta\sigma_{xx}$ (inset of Fig. 3c). This linear-in-B relation is against the normal Shubnikov-de Haas (SdH) oscillations evenly distributed in $1/B$ (see Supplementary Fig. 7) detected in ZrTe₅ with higher carrier density[17,36] and also different from the logarithmic oscillations[37]. As shown in Fig. 3d, we index the oscillations by integers, which is found to be well fitted by $n \sim \frac{C_0}{B} + C_1 * B$, where the $C_0$ term represents contribution from normal $1/B$, and $C_1$ term represents the additional contribution from linear-in-B. The same fitting is also employed to fit fan diagrams obtained from $\Delta\sigma_{xx}$ (Fig. 3e, f), where the linear-in-B trend is highlighted by the shadowed area. Interestingly, one previous experiment measured on polycrystalline ZrTe₅ already exhibits some clues of linear-in-B behaviors[38]. As shown in Fig. 3g, h, we also tilt the magnetic field in **ba** and **bc** planes (defined as $\theta, \phi$), and find the oscillations gradually shift towards high magnetic fields (see Supplementary Fig. 6 for more details on the background subtraction). As summarized in Fig. 3i, j, the angular dependence of a typical peak ($B^* = 0.7$ T) shows anisotropies $\frac{B^*(a)}{B^*(b)} \sim 9$ and $\frac{B^*(c)}{B^*(b)} \sim 7$, respectively, infers a less anisotropic dispersion than anisotropic ratios 13, 8 in a previous report[17].

The two observations drive us to focus on the specific energy dispersion in the magnetic field. As we know, the Landau levels of the Dirac band, in **ac** plane of ZrTe₅, are expressed as: $E_n = \sqrt{2nBv_xv_ye\hbar}$, where $e$ is the elemental charge. With fixed Fermi energy, we get a relation of $n \sim \frac{1}{B}$. However, the Zeeman splitting $\frac{\bar{g}}{2}\mu_B B$ is large (for simplicity, we here use averaged g-factor $\bar{g} \sim 15$[20,39]), results in a modified Landau level dispersion for the valence band: $E_{n(+)} = -\sqrt{2nBv_xv_ye\hbar} + \frac{\bar{g}}{2}\mu_B B$. Now, we get $n \sim \frac{C_0}{B} + C_1 * B$, shows that the Zeeman effect is crucial for the specific n-B relation observed in this work. Due to the high Fermi velocity $v_F \sim 5 \times 10^5$ m/s[21], the energy scale of the kinetic part ($\sqrt{2nBv_xv_ye\hbar}$) is 18 meV*$\sqrt{nB}$, and the Zeeman splitting part ($\frac{\bar{g}}{2}\mu_B B$) is 0.6 meV*B, then the $C_1$ is usually very small and

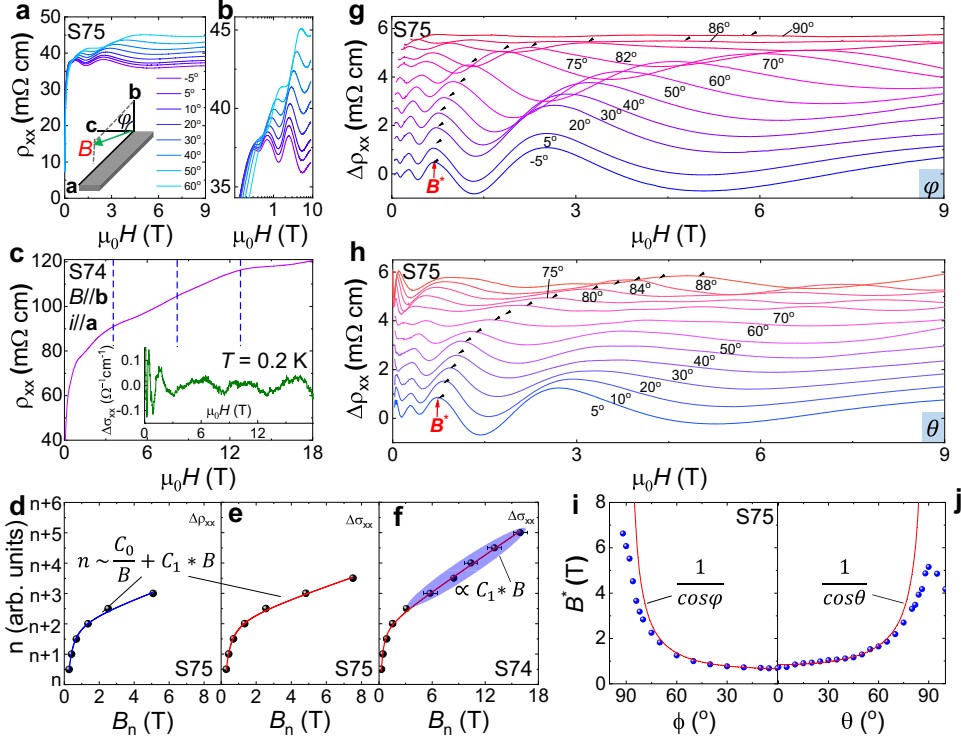

**Fig. 3 | Anomalous resistance oscillations beyond the quantum limit in ZrTe$_5$.**
**a, b** Magnetoresistance measured at different angles with standard and logarithmic scales in sample S75, respectively. The inset shows the experimental setup for rotation. **c** Magnetoresistance measured up to 18 T at 0.2 K in sample S74. The inset shows conductivity oscillations $\Delta\sigma_{xx}$ obtained from background subtraction. **d–f** Landau fan diagrams with arbitrary integers based on oscillatory components $\Delta\sigma_{xx}$ and $\Delta\rho_{xx}$, respectively. Solid blue and red lines are the fittings by the relation

$n \sim \frac{C_0}{B} + C_1 * B$. The shaded blue area in **f** exhibits the linear portion at high magnetic fields, and the error bars are estimated as the broadness (uncertainty) of the peaks and valleys in the inset of **c**. **g, h** Oscillatory component $\Delta\rho_{xx}$ of tilted angles $\theta$ (**bc** plane) and $\phi$ (**ba** plane), respectively. $B^*$ denotes the characteristic peak ($B^* = 0.7$ T). **i, j** The $\theta$ and $\phi$ dependence of characteristic peak $B^*$. Red lines are $1/\cos\theta,\phi$ relations.

overcome by a parabolic band mixing effect for non-ideal Dirac dispersion[40], which is the formal explanation for aperiodic quantum oscillation on higher LLs. However, the appearance of oscillations beyond the quantum limit indicates that the kinetic part should be heavily quenched in the magnetic field. Otherwise, oscillations can only appear at unreal magnetic fields, as the related theory is proposed and discussed in the work[41]. Quenched kinetic energy, or equivalently flattening the band, in Dirac dispersion is parameterized by the enhanced mass and reduced Fermi velocity, which is already indicated by the reduced anisotropy $\frac{B^*(\boldsymbol{a})}{B^*(\boldsymbol{b})}$.

## Field-induced mass enhancement and formation of a topological flat band

In the following, we will show that the kinetic energy of Dirac fermions in ZrTe$_5$ is heavily quenched in magnetic fields. Figure 4a shows the temperature-dependent oscillatory component $\Delta\rho_{xx}$ in sample S74. One prominent behavior is the high-field oscillations dampen quicker than the low-field ones. By extracting the amplitudes of peaks and valleys at characteristic fields, as shown in Fig. 4b, the temperature dependence of amplitudes $\Delta\rho_{xx}$ can be well fitted by the temperature-damping prefactor $\lambda = \frac{2\pi^2 k_B m_c}{\hbar eB} T$ of the Lifshitz–Kosevich (L–K) formula[42], where $k_B$ is the Boltzmann constant (see Supplementary Fig. 9 for more details on the L–K formula fittings). As shown in Fig. 4c, the resulting cyclotron mass ($m_c$) gets heavily enhanced (ratio ~$10^2$) in magnetic fields. The temperature-damping prefactor $\lambda_D$ of the L–K formula for Dirac fermion[43] is written as $\lambda_D = \frac{2\pi^2 k_B |\mu|}{\hbar eB \upsilon_F^2} T$, where the $\mu$ is the chemical potential. Then,

the cyclotron mass $m_c$ fitted by $\lambda$ effectively reflects the quantity $|\mu|/\upsilon_F^2$ in the Dirac system, this is consistent with the universal definition $m_c = \frac{\hbar^2}{2\pi}\frac{\partial A_k}{\partial E}$, where $A_k$ is the extremal area of orbital. By using the fixed carrier density ($p$) constrain in real systems, we come to $m_c = \hbar(6\pi^2 p)^{\frac{1}{3}}/\upsilon_F$, means enhanced $m_c$ corresponds to a reduction of $\upsilon_F$, supporting the formation of a topological flat band. Figure 4d shows the magnetic field dependence of $\upsilon_F$ (zero-field $\upsilon_F$ ~$5\times10^5$ m/s), and the lowest value of $\upsilon_F$ is around $10^3$ m/s at ~20 T. These effects of cyclotron mass enhancement and $\upsilon_F$ reduction are similar to that in twisted bilayer graphene[4], while much stronger than those observed in the NLSM[44] and Kondo insulator[45].

Let's come to a picture based on the above experimental observations. As illustrated in Fig. 5a, the topmost Rashba-splitted band A flattens in a magnetic field, and the kinetic energy is heavily quenched. At the same time, the lower band is still dispersive with large kinetic energy and then goes up quickly in the magnetic field. The simplified band A, as a topological flat band, thus exhibits quenched kinetic energy and dominated Zeeman energy. Then, the Landau levels will bend and reappear to FL, which is essentially different from the normal quantum oscillations. The quantum limit defined in this picture is equal to that in normal quantum oscillations estimated by carrier concentration, namely the critical field where the LLL solely occupies (higher LLs will bend over and reappear later). We plot the kinetic energy $E_{LL,kinetic} = -\sqrt{2nB\upsilon_x\upsilon_y e\hbar}$ (Fig. 5b) with fitted $m_c$, which quenches at small fields. As shown in Fig. 5c, we plot the total energy $E_n = -\sqrt{2nB\upsilon_x\upsilon_y e\hbar} + \frac{\tilde{g}}{2}\mu_B B$ ($\tilde{g}$ ~15), and find the Landau levels cross the FL, distributing evenly in high magnetic fields. Therefore, the anomalous resistance oscillations observed

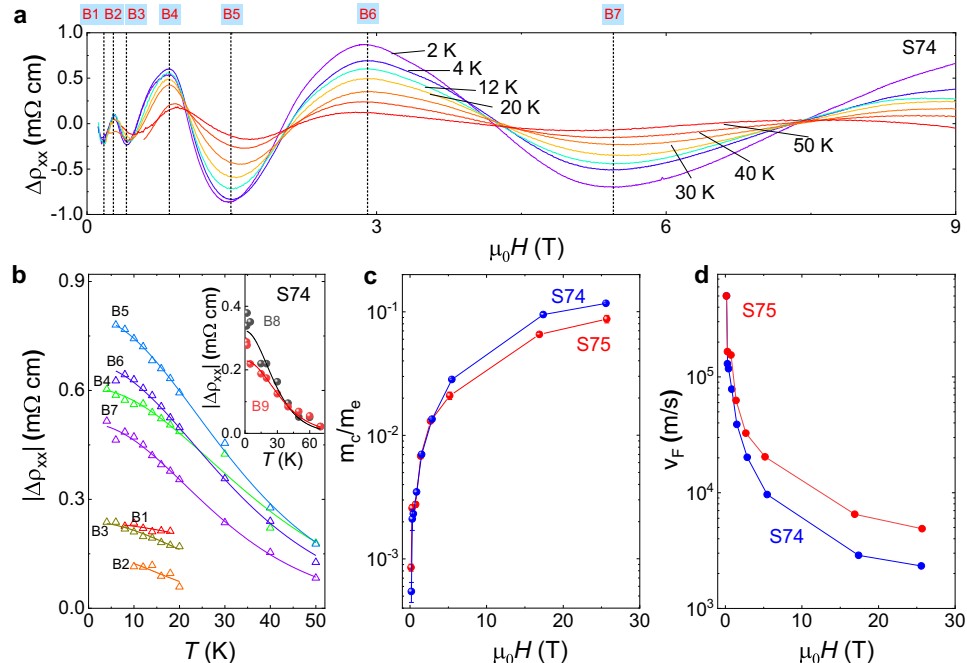

**Fig. 4 | A topological flat band evidenced by field-induced mass enhancement in ZrTe₅.** **a** Temperature-dependent oscillatory component $\Delta\rho_{xx}$ of sample S74. B1–B7 denotes the seven characteristic peaks and valleys that are subjected to the L–K formula fitting. **b** Temperature-dependent amplitudes of the component $\Delta\rho_{xx}$ fitted by the L–K formula. By a narrower fitting window, clear temperature dependence of B1, B2, and B3 can be accessed (Supplementary Fig. 10). The inset shows the L–K formula fitting at fields higher than 9 T measured in a 33 T water-cooled magnet. **c, d** Enhanced cyclotron mass ($m_c$) and reduced Fermi velocity ($v_F$) in the magnetic fields. Error bars are estimated from the standard deviations in the fittings of temperature-damping prefactor $\lambda_D$ of the L–K formula.

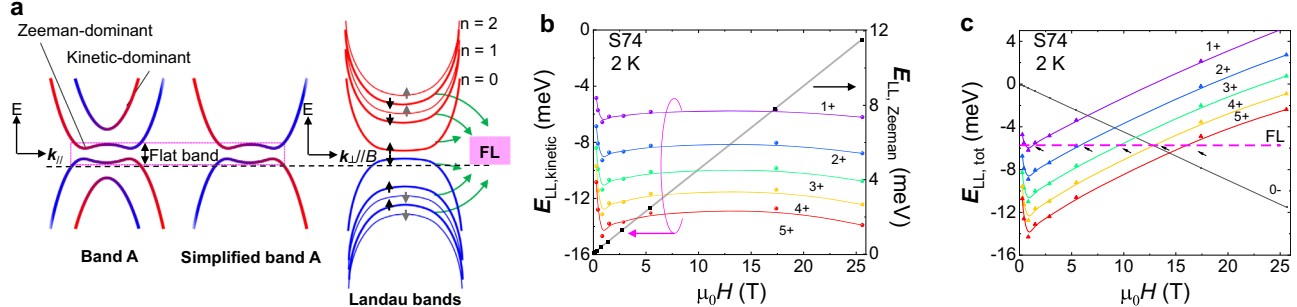

**Fig. 5 | Quenched kinetic energy of Landau levels.** **a** Formation of Landau levels on the topological flat band with dominant Zeeman effect. **b** The kinetic energy, $E_{LL,kinetic} = -\sqrt{2n B v_x v_y e\hbar}$, of Landau levels obtained from experimental cyclotron mass, showing the quenched kinetic energy at high fields. **c** The total energy of Landau levels, including the dominant Zeeman effect $\frac{g}{2}\mu_B B$, exhibiting the reappearance of Landau levels across the Fermi level (FL) denoted as a dashed pink line.

beyond the quantum limit are consistent with forming a topological flat band in ZrTe₅.

## Discussion

In conclusion, our work demonstrates that the 3D Dirac material ZrTe₅ can be naturally transformed into a topological flat band system by symmetry breaking at low temperatures without invoking van der Waals heterostructure-based engineering. In this topological flat band, we observe anomalous resistance oscillations beyond the quantum limit, originating from the quenched kinetic energy of electrons on this flat band. Our work has two consequences: the first one is that the intrinsically formed topological flat band provides another route to create a topological flat band system beyond van der Waals heterostructures-based engineering; the second one is that the realization of a topological flat band in a 3D archetypical Dirac system opens the door for studying the specific exotic phenomena for topologically correlated effects uniquely existed in the dimensionality of three.

## Methods

### Crystal growth and characterizations

For single crystals of ZrTe₅ grown by the Te-flux method: Starting materials Zr (Alfa, 99.95%) and Te (99.9999%) were sealed in a quartz tube with a ratio of 1:300 and placed in a box furnace. Then, the materials were heated up to 900 °C, followed by shaking the melt, and kept for two days. The temperature was then lowered to 660 °C within 7 hours and cooled to 460 °C in 200 hours. Single crystals of ZrTe₅ were isolated from Te-flux by centrifuging at 460 °C. Iterative temperature cycling was adopted to increase the size of ZrTe₅.

For single crystals of ZrTe₅ grown by CVT method:ZrTe₅ powder was synthesized by stoichiometric Zr (aladdin, 99.5%) and Te (99.999%). ZrTe₅ powder and iodine were sealed in a quartz tube and then placed in a two-zone furnace with a temperature gradient of 540 °C and 445 °C for one month.

The crystal structure and composition were checked by powder x-ray diffraction (XRD, Rigaku) and energy dispersive x-ray (Hitachi-

SU5000) (see Supplementary Fig. 1 for details). Crystalline structure was illustrated by the VESTA software[46].

## Transport measurements

The sample must be cleaved to be ribbon-like, forcing the current flow to be evenly distributed. All the surfaces exposed to air should be cut or cleaved to avoid Te-flux contamination. Due to the reaction between silver paste and $ZrTe_5$, the usual bonding method with silver paste directly applied on the sample surface is not applicable here, which causes huge contact resistance and contaminates the electrical transport. To make good contacts, the surface was cleaned first by Ar plasma, 5 nm Ti/50 nm Au was then deposited on the *ac* surface with a homemade Hall bar mask. Then, silver paste was used to bond the Au wires for typical four-probe electrical contacts. The contact resistance is around several Ohms, and no sizable increase in contact resistance was observed over several months.

Electrical transport measurements above 2 K were carried out in a Quantum Design Physical Property Measurement System 9 T (PPMS-9 T) with a sample rotator. Measurements below 2 K were done in a top-loading dilution fridge (Oxford TLM, base temperature ~20 mK). High-field Measurements were carried out in a water-cooled magnet with steady fields up to 33 T in the Chinese High Magnetic Field Laboratory (CHMFL), Hefei. Lock-ins (SR830) were used to measure the resistance and the $2\omega$ resistance with a Keithley 6221 AC/DC current source.

## Torque magnetometry

Magnetic torque measurements were carried out on a piezoresistive cantilever with a compensated Wheatstone bridge. The tiny sample cut from the pristine crystal was mounted on the tip of the cantilever. The resulted torque is roughly calculated by the relation: $\tau = \frac{4}{3} \frac{at^2}{\pi_L} \frac{\Delta V}{iR_s}$, where a is the leg width, t is the leg thickness, coefficient $\pi_L = 4.5 \times 10^{-10} m^2/N$, $\Delta V$ is the voltage drop on the Wheatstone resistance bridge, *i* is the current fed into the bridge and $R_s$ is the resistance of sample leg.

## Data availability

The data generated in this study are provided in the Source Data file. Additional data related to the current study are available from the corresponding author upon request. Source data are provided with this paper.

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

## Acknowledgements

We thank Shuang Jia for providing cantilevers and supporting us on the initial construction of the torque setup. We thank Shiliang Li and Jin Ding for assisting with the Laue measurement. D. Xing thanks Jianghao Jin, Yuxin Yang, and Tianping Ying for assisting with crystal growth and Jianfei Xiao for helping with contact fabrication. C.-L. Zhang was supported by the National Key R&D Program of China (Grant No. 2023YFA1407400) and a start-up grant from the Institute of Physics, Chinese Academy of Sciences. J. Luo was supported by the National Science Foundation of China (Grants No. 12134018). J. Zhang was supported by the National Key R&D Program of the MOST of China (Grant No. 2022YFA1602602) and the Natural Science Foundation of China (Grant No. 12122411). A portion of this work was carried out at the Synergetic Extreme Condition User Facility (SECUF).

## Author contributions

C.-L.Z. conceived and supervised the project. D.X., S.P., B.T., Z.W., J.Z., J.L., and C.-L.Z. performed the electrical transport experiments; C.-L.Z. did the magnetic torque experiments. D.X. and C.-L.Z. grew the single-crystalline samples and characterized them by XRD and EDX; C.-L.Z., and D.X. analyzed the data. C.-L.Z. wrote the paper with input from all other authors.

## Competing interests

The authors declare no competing interests.
