## [Peer Review File · Nature Communications]

Rashba-splitting-induced topological flat band detected by anomalous resistance oscillations beyond the quantum limit in ZrTe₅Reviewers' Comments:

Reviewer #1:

Remarks to the Author:

The electronic structure of ZrTe₅ is a hot topic now. There are plenty of models to describe different experimental data. In the paper of Dong Xing et al., based on the authors' experiments, it is proposed that ZrTe₅ can host a flat band. The idea is exciting. Meanwhile, there are questions about oscillations analysis:

Question 1: The crucial point of deriving the formulas for n is the assumption about fixed Fermi energy (line 179).

What fixes the Fermi energy?

There is a tiny Fermi surface hole pocket, and the magnetic field is near the ultra-quantum regime. Under these conditions, the number of holes, not the Fermi energy, should be fixed. The Fermi energy becomes a function of the magnetic field, and the expression for n (lines 182-183) should be modified. It is important to explain why the Fermi energy is fixed.

Question 2: The ultra-quantum limit refers to a situation when all charge carriers (holes) are confined to the lowest Landau level (LL).

In the paper, it is written that there are oscillations beyond the "ultra-quantum limit" field (the title, the abstract, lines 160-162). Nevertheless, further, it is shown that these oscillations correspond to LL crossings. So, there are a lot of occupied hole LLs for the magnetic field equal to and above the field, called the "ultra-quantum limit."

When the authors write about "oscillations beyond ultra-quantum limit," this actually means that the oscillations are still below the correct ultra-quantum limit but beyond some wrongly estimated "ultra-quantum limit". This looks like a recipe to add loud words in the title for nothing.

This mention of the "quantum limit" is misleading because a reader expects that the oscillations beyond the quantum limit are not related to LLs crossings, see, i.e.,

S. Galeski et al., "Signatures of a magnetic-field-induced

Lifshitz transition in the ultra-quantum limit of the topological semimetal ZrTe₅" Nat. Commun. 13, 7418 (2022).

where the actual ultra-quantum limit magnetic field was reached.

Question 3: The paper's results predict that the Landau index n increases with the magnetic field; see Fig.3d-f. Then, the oscillations will be present in an arbitrary high magnetic field. However, this conclusion contradicts the experimental results of S. Galeski et al. (2022), where the real ultra-quantum limit was reached.

Additionally, if the number of occupied LLs increases with the magnetic field, and the Fermi energy is fixed, then the number of holes skyrockets to infinity. From where will all these holes come?

Question 4: How can the paper's results be affected by the assumption that ZrTe₅ is a multivalley semimetal?

(i.e., as in

Zoltán Kovács-Krausz et al., Revealing the band structure of ZrTe₅ using multicarrier transport, Phy.Rev.B 107, 075152 (2023)

The Landau index n is also not precisely proportional to $1/B$, but another explanation was given.)

Question 5: The authors of this paper plot their data in magnetic field B . Usually, in other papers, i.e., as mentioned above, SdH data and fan diagrams are plotted as a function of the inverse field $1/B$. Can the authors give a comparison of their data with the previous ones?

Reviewer #2:

Remarks to the Author:

Please find the attached review report file.

The manuscript “Rashba-splitting-induced topological flat band detected by anomalous resistance oscillations beyond the quantum limit in ZrTe5” by D. Xing et al. is interesting and well-written. The authors conducted angular and temperature-dependent magnetotransport measurements to demonstrate that ZrTe5 possesses a topological flat band. The authors provided minor details to support their claim. The overall quality of the research is good, and the data are of high quality. This work will be crucial for understanding topological/Dirac materials and compounds with flat band. However, I encountered some issues with how the authors analyzed their data.

- For example, the torque signal appears to exhibit periodic behavior (Fig. 2(a)) that can be described by any periodic sine or cosine functions. The authors added an additional term, $A_2 \sin^2(\theta)$, to fit the data. I am unsure about the fundamental physics behind this. While the authors later connected the A_2 term to draw important conclusions, it might be beneficial for them to provide a clearer explanation.
- Similarly, the most important aspect of this work is to demonstrate that the kinetic energy (or velocity) quenches to exhibit flat band behavior. The authors used LK fit analyses for this purpose (Fig. 4). I am a little concerned about the analyses for low field data (B1, B2, B3) where the oscillations are barely visible and do not seem to show temperature dependence. Additionally, the LK fit for a few points does not yield any meaningful values for the effective mass. Furthermore, the authors have shown that the effective mass depends on the magnetic field (Fig. 4(c)). What could be the physical meaning of this observation?

Reviewer #3:

Remarks to the Author:

The manuscript titled 'Rashba-splitting-induced topological flat band detected by anomalous resistance oscillations beyond the quantum limit in ZrTe5' by Dong Xing et al present the the polar distortion and Rashba splitting on the band detected by torque magnetometry and the anomalous Hall effect. SdH oscillations are shown to be periodic in B instead of the usual $1/B$ in high magnetic fields, with an unusually high enhancement in the cyclotron mass of $\sim 10^2$ times at field ~ 20 T.

The ZrTe5 shows quite interesting and varied properties at low temperature, depending on the crystal quality. There is a lot literature on the crystals with extra Tellurium, with dominating n-type carrier concentration. This study is on a clean ZrTe5 crystal, and present quite interesting results. The results presented in the manuscript are of good quality. However I have following reservations before I recommend it for the publication in this journal. Author should address these points.

1. It is important to add a image, description (including Laue Diffraction pattern) of the ZrTe5 crystal morphology and its chemical composition at least in the supplementary information. Since this study pertains to the angular field dependent results, geometrical factors owing to crystal shape might introduce an error in the measurement.
2. Authors have discussed the anomalous Hall resistivity in the compound, and discussed the carrier concentration obtained from the high field data using single band model. However, the magnetic field range used for the calculation of Hall coefficient is not clear, and whether it is the same for all temperatures.
3. The value of carrier concentration ($n \sim 10^{14} \text{ cm}^{-3}$) obtained from Hall resistivity is too low, and it is not apt to compare it with the 2 dimensional density of carrier. I would like to see the proper fitting of the data and calculation for at least two temperatures.
4. What was the thickness of the sample used for the Hall study and how was it measured?
5. ZrTe5 is a non-magnetic compound, what is the reason for Anomalous Hall resistivity? Are the authors aware of any other similar report on the compound?
6. There have been some reports on ZrTe5, where SdH oscillations periodic in B (PRB 98, 165119, 2018), and $\log B$ (Sci. Adv. 4, eaau5096 (2018)) are discussed. Author should compare the cyclotron mass with these reports, and discuss the variation.
7. How does the plausibility of the topological flat band in the material makes it a 3D counter part of graphene which has Dirac cone shape band at Fermi level?

Authors' reply to Reviewers' comments

Reviewer #1 (Remarks to the authors): The electronic structure of ZrTe₅ is a hot topic now. There are plenty of models to describe different experimental data. In the paper of Dong Xing et al., based on the authors' experiments, it is proposed that ZrTe₅ can host a flat band. The idea is exciting.

Authors: We thank the Reviewer #1 for thinking the observed 'flat band' in ZrTe₅ is 'exciting', which is the main claim based on combined experimental results in our work.

Reviewer #1: The crucial point of deriving the formulas for n is the assumption about fixed Fermi energy (line 179). What fixes the Fermi energy? There is a tiny Fermi surface hole pocket, and the magnetic field is near the ultra-quantum regime. Under these conditions, the number of holes, not the Fermi energy, should be fixed. The Fermi energy becomes a function of the magnetic field, and the expression for n (lines 182-183) should be modified. It is important to explain why the Fermi energy is fixed.

Authors: We thank the Reviewer #1 for raising the question of 'fixed carrier density or fixed Fermi energy', which is a point should be discussed. The constraint of fixed carrier density is adopted for normal Shubnikov–de Haas (SdH) oscillations in the quantum limit where only the last Landau level (LLL) occupies. The Fermi energy follows the LLL, indicating the Fermi energy is not fixed in the quantum limit. **However, our work reports the observation of anomalous SdH oscillations beyond this normal picture.** For this new picture, the constraint of fixed Fermi energy is reasonable due to the reappearance of LLs in the quantum limit. Furthermore, whether fixed Fermi energy or carrier density is adopted will not alter the linear-in-B relation. Here, we explain the distinction between the normal SdH and the anomalous one in detail. We adopt a gapless Dirac model for simplicity. Details with mass term, LLLs and Fermi energy evolution can be found in related work (PRB 102, 041204 (2020); Ref. [41] in the main text):

Fig. R1 The quantum oscillations in two Dirac models. **a**, Typical dispersion of Landau levels (LL) in a pure gapless Dirac model, signifies the square root field dependence. The LLL is pinned to zero energy, with the quantum limit defined as the critical field of depopulation of $n = 1$ LL. **b**, The dispersion of LLs in a gapless Dirac model with giant Zeeman effect, the inverted LLs show quantum oscillations beyond the quantum limit. The LLLs are not shown here; the detailed analysis with LLLs can be found in Ref. [41] of the main text.

- A. **The SdH in a pure Dirac model.** As shown in Fig. R1a, the dispersion of LLs of a gapless Dirac model exhibits $\sim \sqrt{B}$ relation. The field-dependent Fermi energy before the quantum limit is sawtooth-like, which is usually simplified by a fixed Fermi energy as indicated by the green solid line. However, when the system enters the quantum limit where only the LLL occupies, the Fermi energy bends down to follow the LLL under the constraint of conserved carrier. For normal SdH, the Fermi energy is not fixed in the quantum limit. The normal SdH oscillates in $1/B$, and the quantum limit can be estimated by the extremal orbit related to carrier density, which is the definition we adopted in our work.
- B. **The SdH in a modified Dirac model.** As discussed in the main text, a typical Dirac material's energy scale of cyclotron motion is much larger than the Zeeman effect due to the high Dirac velocity $\sim 10^5$ m/s. Therefore, it is challenging to find a Dirac material where the Zeeman effect dominates over the cyclotron motion. However, our experiments show that the ZrTe₅ with polarity exhibits quenched Fermi velocity, and non- $1/B$ SdH appears in the quantum limit. Therefore, the LLs **MUST** (experimental parameters fully nail down the dispersions of LLs) follow the inverted behavior as shown in Fig. R1b (=Fig. 5a in the main text). Compared to the pure Dirac model, we see the fixed Fermi energy is now a good approximation in the quantum limit, because lots of LLs populate and depopulate over the Fermi energy, like the normal SdH on higher LLs.
- C. **Detailed analytical results and simulations based on a modified Dirac model.** In addition to the simple argument mentioned above, Ref. [41] carried out systematic simulations based on 2D and 3D Dirac models. As shown in Fig. 1 & 2 of Ref. [41], The anomalous oscillations are caused by the inverted LLs reappearing in the quantum limit. The definition of quantum limit is the same as ours. The simulations with the fixed energy and fixed carrier are both valid due to the inversion of LLs. As shown in Fig. 1(c) and 2(c) of Ref. [41], both results follow the linear-in-B relation with a slightly different period. Therefore, the fixed Fermi energy constraint is reasonable in our work. We added several sentences on this point in the main text.

Reviewer #1: The ultra-quantum limit refers to a situation when all charge carriers (holes) are confined to the lowest Landau level (LL). In the paper, it is written that there are oscillations beyond the "ultra-quantum limit" field (the title, the abstract, lines 160-162). Nevertheless, further, it is shown that these oscillations correspond to LL crossings. So, there are a lot of occupied hole LLs for the magnetic field equal to and above the field, called the "ultra-quantum limit." When the authors write about "oscillations beyond ultra-quantum limit," this actually means that the oscillations are still below the correct ultra-quantum limit but beyond some wrongly estimated "ultra-quantum limit". This looks like a recipe to add loud words in the title for nothing. This mention of the "quantum limit" is misleading because a reader expects that the oscillations beyond the quantum limit are not related to LLs crossings, see, i.e., S. Galeski et al., "Signatures of a magnetic-field-induced Lifshitz transition in the ultra-quantum limit of the topological semimetal ZrTe₅" Nat. Commun. 13, 7418 (2022). where the actual ultra-quantum limit magnetic field was reached.

Authors: We thank the Reviewer #1 for this related question. This question is highly associated with the new mechanism illustrated in Fig. R1b, and the definition of quantum limit is also explained in the first reply to the Reviewer #1. One note is that: compared with the low carrier density of our samples ($10^{14}\sim 10^{15}$ cm⁻³) with no peak on the resistivity versus temperature curve, the experiment [S. Galeski et al., Nat. Commun. 13, 7418 (2022)] was carried out on samples with higher carrier density ($10^{16}\sim 10^{17}$ cm⁻³), indicating the quantum limit in our sample must be lower. One can also immediately judge this argument by looking at raw data of the resistivity of two classes of flux-grown samples. **The Hall and longitudinal resistivity are much higher in samples with no peak on resistivity versus temperature (RT) curve, indicating a much insulating bulk. Therefore, without being fitted by the Drude model, the raw resistivity data already tell us that the quantum limit in our samples must be lower than the ones in S. Galeski et al. (2022) of typically ~ 1 T.**

Reviewer #1: The paper's results predict that the Landau index n increases with the magnetic field; see Fig.3d-f. Then, the oscillations will be present in an arbitrary high magnetic field. However, this conclusion contradicts the experimental results of S. Galeski et al. (2022), where the real ultra-quantum limit was reached. Additionally, if the number of occupied LLs increases with the magnetic field, and the Fermi energy is fixed, then the number of holes skyrockets to infinity. From where will all these holes come?

Authors: We thank the Reviewer #1 for this related comment. This question is also

answered in the first reply to the Reviewer #1. We are using the original LLs' numberings because the LLs with original numberings will bend over and cross the Fermi energy again in inverted order, then the numbering increases, and the carrier density is always conserved. Our numbering method is consistent with work Ref. [41].

Reviewer #1: How can the paper's results be affected by the assumption that ZrTe₅ is a multivalley semimetal? (i.e., as in Zoltán Kovács-Krausz et al., Revealing the band structure of ZrTe₅ using multicarrier transport, *Phy.Rev.B* 107, 075152 (2023) The Landau index n is also not precisely proportional to $1/B$, but another explanation was given.)

Authors: We thank the Reviewer #1 for this critical question. The work [*Phy.Rev.B* 107, 075152 (2023)] is based on a CVT sample, which is a well-established multicarrier system. Now, the research field is reaching a consensus that the flux system is a single carrier system at low temperatures [*Nat. Commun.* 12, 406 (2021); *Nature* 569, 537-541 (2019); arXiv:2101.02681 (2021)]. When the Fermi level is elevated in a multi-carrier system, and the increased free carrier will screen the polar order, smearing the Rashba splitting (see Fig. S3 of SI), then the physics discussed in our work will disappear. The Reviewer #1 pointed out that Fig. S3 in the SI of work [*Phys. Rev. B* 107, 075152 (2023)] shows the Landau fan diagram (n versus $1/B$) deviates from linearity at a high magnetic field **before the quantum limit**, which is already well-explained by Ando's work [*Phys. Rev. B* 84, 035301 (2011)]. We added work [S. Galeski et al., *Nat. Commun.* 13, 7418 (2022)] to the references to distinguish the anomalous quantum oscillations in our work.

Reviewer #1: The authors of this paper plot their data in magnetic field B . Usually, in other papers, i.e., as mentioned above, SdH data and fan diagrams are plotted as a function of the inverse field $1/B$. Can the authors give a comparison of their data with the previous ones?

Authors: We thank the reviewer for this suggestion. As shown in Fig. R2, the oscillations are unevenly distributed and exhibit essentially different behavior compared with ones observed on samples with higher carrier density [S. Galeski et al., *Nat. Commun.* 13, 7418 (2022)].

Fig. R2 a, The quantum oscillations replotted in $1/B$ and **b**, Landau fan diagram.

Reviewer #2 (Remarks to the authors): The manuscript “Rashba-splitting-induced topological flat band detected by anomalous resistance oscillations beyond the quantum limit in ZrTe5” by D. Xing et al. is interesting and well-written. The authors conducted angular and temperature-dependent magnetotransport measurements to demonstrate that ZrTe5 possesses a topological flat band. The authors provided minor details to support their claim. The overall quality of the research is good, and the data are of high quality. This work will be crucial for understanding topological/Dirac materials and compounds with flat band.

Authors: We thank the Reviewer #2 for these critical suggestions. We also think comments like ‘*data are of high quality*’ and ‘*this work will be crucial for understanding topological/Dirac materials and compounds with flat band*’ from the Reviewer #2 are objective. The reason is that the sample growth method and quality cause disputes in this field. Our work brings another view of accessing the flat band in topological materials.

Reviewer #2: For example, the torque signal appears to exhibit periodic behavior (Fig. 2(a)) that can be described by any periodic sine or cosine functions. The authors added an additional term, $A_2 \sin^2(\theta)$, to fit the data. I am unsure about the fundamental physics behind this. While the authors later connected the A_2 term to draw important conclusions, it might be beneficial for them to provide a clearer explanation.

Authors: We thank the Reviewer #2 for this fundamental question about A_2 term, which induces the flat band in ZrTe5. We will explain the technique and related analyses in detail:

- A. Torque magnetometry is used to measure the anisotropy of magnetization. It prevails in detecting the de Haas-van Alphen (dHvA) effect for mapping the Fermi surface. Another important application of the torque technique is to detect the spontaneous symmetry breaking in quantum materials, like the nematic phase in iron-based superconductors or order parameter-related physics in other systems. The symmetry breaking on the electronic or lattice structure will modify the magnetic susceptibility χ tensor, which can be detected by the torque signal $\tau = \mu_0 V \mathbf{M} \times \mathbf{H}$. As for highly symmetric crystals without anisotropy on \mathbf{M} , the torque signal is always zero. For example, if ZrTe5 is tetragonal with C4 rotation symmetry along \mathbf{a} -axis, then $\chi_{cc} = \chi_{bb} = 0$, the A_1 term is zero. However, if the crystal breaks the C4 symmetry at a critical temperature, where the torque signal will abruptly appear. This is the main logic we adopted for detecting the symmetry breaking by ultra-sensitive torque.
- B. Now, we turn to the details of terms A_1 and A_2 . As shown in Fig. 2a in the main text, the torque is periodic in π at all measured temperatures, which means that the A_1

term directly from the orthorhombic structure (room temperature structure of ZrTe₅) mainly contributes to the signal. However, we unexpectedly find that the raw signal shows a difference between amplitudes of the two peaks around 45 deg and 135 deg at low temperatures, as shown in Fig. R3. The data can be fitted with including the A₂ term $\sim \sin^2 \theta$, as shown in the fitted curve in Fig. 2a in the main text.

- C. The appearance of A₂ term indicates the symmetry of ZrTe₅ breaks into a lower symmetric structure compared with the pristine orthorhombic one. However, as explained in SI, we still cannot nail down the specific structure by torque, although we have provided details of the possible lower symmetric structures. Therefore, we further performed the nonlinear transport and found that the ZrTe₅ develops ferroelectric polarity along **b**-axis. Based on the two experiments, we can conclude that the low-temperature structure of ZrTe₅ adopted a non-orthorhombic polar structure. The exact structure needs further ultrasensitive spectroscopic development. Related discussions have been updated in the main text.

Fig. R3: Torque signals measured at 270 K and 4 K, respectively. The dash lines are fittings.

Reviewer #2: Similarly, the most important aspect of this work is to demonstrate that the kinetic energy (or velocity) quenches to exhibit flat band behavior. The authors used LK fit analyses for this purpose (Fig. 4). I am a little concerned about the analyses for low field data (B1, B2, B3) where the oscillations are barely visible and do not seem to show temperature dependence. Additionally, the LK fit for a few points does not yield any meaningful values for the effective mass. Furthermore, the authors have shown that the effective mass depends on the magnetic field (Fig. 4(c)). What could be the physical meaning of this observation?

Authors: We thank the Reviewer #2 for this comment. **1.** Due to the presentation, we now locally amplify Fig. 4a, as shown in Fig. R4, temperature-dependent B1, B2, B3 can be resolved (B1 and B3 are better than B2). **2.** The quality of high-field data for samples

S74 and S75 is different due to the difficulty in subtracting the background of high-field data of S74. As the Reviewer #2 pointed out, only three points for B8 and B9 in the inset of Fig. 4b are not so meaningful for L-K fitting. However, we have measured another sample in the high-field facility for reproducibility (Fig. R5). The L-K fitting of B7 and B8 with 5 data points in sample S75 is reasonable and produces high-field effective mass similar to that of S74. Therefore, the effective mass presented in Fig. 4c is reasonable.

Fig. R4: Locally amplified of Fig. 4a in the main text.

For the effect of field-induced enhancement of effective mass: As shown in the illustration in Fig. 2d and 5a, the magnetic field flattens the band around the Fermi level by gapping out the Rashba crossing, causing the enhanced effective mass. Therefore, the kinetic energy of electrons is quenched, causing anomalous quantum oscillations dominated by the Zeeman effect. The physical meaning is that a small Zeeman magnetic field in the polar system with dilute carriers can realize the topological flat band.

Fig. R5: L-K fitting at low and high fields in another sample S75.

Reviewer #3 (Remarks to the authors): The manuscript titled ‘Rashba-splitting-induced topological flat band detected by anomalous resistance oscillations beyond the quantum limit in ZrTe5’ by Dong Xing et al present the the polar distortion and Rashba splitting on the band detected by torque magnetometry and the anomalous Hall effect. SdH oscillations are shown to be periodic in B instead of the usual 1/B in high magnetic fields, with an unusually high enhancement in the cyclotron mass of $\sim 10^2$ times at field ~ 20 T. The ZrTe5 shows quite interesting and varied properties at low temperature, depending on the crystal quality. There is a lot literature on the crystals with extra Tellurium, with dominating n-type carrier concentration. This study is on a clean ZrTe5 crystal, and present quite interesting results. The results presented in the manuscript are of good quality.

Authors: We thank the Reviewer #3 for the comments ‘*This study is on a clean ZrTe5 crystal, and present quite interesting results. The results presented in the manuscript are of good quality.*’ We agree with the Reviewer #3’s emphasis on the quality of sample and the consequent measured results, because high-quality samples and data will gradually make the society address and distinguish the key properties of ZrTe5 out of many disputes, which is another contribution of our work in addition to observing a topological flat band.

Reviewer #3: It is important to add a image, description (including Laue Diffraction pattern) of the ZrTe5 crystal morphology and its chemical composition at least in the supplementary information. Since this study pertains to the angular field dependent results, geometrical factors owing to crystal shape might introduce an error in the measurement.

Authors: We thank the Reviewer #3 for this helpful suggestion. We now add Fig. S1 (Fig. R6 below) to SI, including a measured **sample image, as-grown crystal image, power XRD, Laue, and EDX data**. As the Reviewer #3 pointed out, the sample geometry is crucial for angular-dependent electrical measurement, especially for the in-plane Hall check in the SI, which requires a well-defined Hall bar structure with reduced thickness. We measured many samples and found following points: **1.** The sample must be cleaved to be ribbon-like, forcing the current flow to be evenly distributed; **2.** all the surfaces explored in the air should be cut or cleaved to avoid the Te flux contamination; **3.** In this **bulk material-based** transport, Ti/Au evaporation and deposition are extremely important for contacts.

Fig. R6: Basic information on crystal morphology and structural characterizations.

Reviewer #3: Authors have discussed the anomalous Hall resistivity in the compound, and discussed the carrier concentration obtained from the high field data using single band model. However, the magnetic field range used for the calculation of Hall coefficient is not clear, and whether it is the same for all temperatures.

Authors: We thank the reviewer for this question. There are two main classes of ZrTe₅ samples: CVT-grown and flux-grown. The CVT samples are always multicarrier due to a higher Fermi level. Researchers have concluded that these flux-grown samples have a single band at low temperatures around the Γ point [Nat. Commun. 12, 406 (2021); Nature 569, 537-541 (2019); arXiv:2101.02681 (2021)], then the Hall data should be fitted by a single Drude model. However, these samples show anomalous Hall (also questioned by the Reviewer #3), which is still under debate; we will answer this point in the next reply. The fitting to the Hall effect depends on how we understand the anomalous Hall effect and how to disentangle the two contributions. Unlike the flux sample (from G. Gu et al.) with a resistivity peak, our sample shows no hole-to-electron transition from 300 to 2 K. Therefore, we adopted a single Drude model to fit the full temperature range (300 K) and magnetic field (9 T). One note is that our interpretation is based on our previous work on the in-plane Hall effect [C.-L. Zhang et al. PNAS, 118 (44) e2111855118 (2021)] and our scheme is also similar to the process adopted by work [J. Mutch et al. Sci. Adv. 5, eaav9771 (2019) & arXiv:2101.02681 (2021)]:

- A. The carrier concentration can be extracted from the ordinary Hall effect with linear-in-B relation with the slope R_H reflects the Hall coefficient, which is usually extracted from the fitting to the ρ_{yx} at small fields where the anomalous term is not large enough.
- B. However, the addition relation cannot be directly applied to the resistivity, so we adopted the conductivity scheme with all raw resistivity tensors converted into conductivity. Then, the anomalous and ordinary terms of Hall conductivity can be safely separated.
- C. As illustrated in work [arXiv:2101.02681 (2021)], we use the Drude model to fit the Hall conductivity in the whole magnetic field range with **the characteristic Drude peak (resonant mobility peak) coinciding with the raw data**.
- D. As we can see from the raw data and fitted curve, the remaining Hall conductivity with the Drude term subtracted is the so-called anomalous Hall conductivity, as plotted in Fig. 2b in the main text (or Fig. R7 below), is almost reached a constant at a higher magnetic field, which makes the two carriers or multicarrier model fitting almost impossible, which is also pointed out by work [arXiv:2101.02681 (2021)].

Reviewer #3: The value of carrier concentration ($n \sim 10^{14} \text{ cm}^{-3}$) obtained from Hall resistivity is too low, and it is not apt to compare it with the 2 dimensional density of carrier. I would like to see the proper fitting of the data and calculation for at least two temperatures.

Authors: We thank the reviewer for this suggestion. Typical thickness is around 50-100 μm , and typical carrier density is around 10^{14} - 10^{15} cm^{-3} for ZrTe₅ samples used in our work. Such low carrier density typically appears in weak topological insulators like the recently known BiBr series [PRB 106, 075206 (2022)]. As shown in Fig. R7, in addition to the 2K fitting shown in Fig. 1d in the main text, we also show the Drude fittings at 18 K and 30 K, respectively. Our results are also consistent with the fitted results from works [J. Mutch et al. Sci. Adv. 5, eaav9771 (2019) & arXiv:2101.02681 (2021)]. As for the 2D limit, we do not think the 2D layer can inherit the ultralow density of bulk due to the band modifications.

Fig. R7: Drude fittings to Hall conductivity of sample S75 at 18 K and 30 K, respectively.

Reviewer #3: What was the thickness of the sample used for the Hall study and how was it measured?

Authors: We thank the reviewer for this question. The thickness of samples S74 and S75 are 46 μm and 105 μm , respectively, measured by microscope with camera.

Reviewer #3: ZrTe5 is a non-magnetic compound, what is the reason for Anomalous Hall resistivity? Are the authors aware of any other similar report on the compound?

Authors: We thank the reviewer for this critical question. This is a puzzle in ZrTe5 research initiated by the publication T. Liang et al., Nat. Phys. 14, 451 (2018). Several following experimental works, including several theoretical proposals, try to address this anomalous Hall in nonmagnetic ZrTe5. Weyl picture is considered initially; later, a picture based on weak topological insulator (WTI) is also discussed [T. Liang et al., Nat. Phys. 14, 451 (2018); arXiv:2101.02681 (2021); npj Quantum Materials 5, 36 (2020)]. However, the consensus on the mechanism of anomalous Hall is still not reached.

Here, we try to address this problem based on our new experimental findings in Fig. 2. **First**, as indicated by Nat. Commun. 12, 406 (2021), our results are also based on a WTI

picture without inducing the Weyl phase. *Second*, we discover a polar axis along *b*-axis, which incurs Rashba splitting in the *ac* plane. *Third*, the anomalous Hall effect disappears in the in-plane configuration. Based on these key observations, we proposed a mechanism as illustrated in Fig. 2d. When the magnetic field is along out-of-plane (*b*-axis), the Zeeman coupling will gap out the Rashba crossing, which will generate the Berry curvature hot spot condensed on the gap regime, contributing the anomalous Hall. This temperature-dependent anomalous Hall conductivity tracks together with the polar distortion, prove the role of Rashba splitting. The most important one is that we cannot observe the in-plane Hall effect in these flux-grown samples, which is consistent with our model. Because the in-plane magnetic field never gaps out the Rashba crossing, making the Berry curvature component Ω_z vanishing, then the anomalous Hall is zero.

Reviewer #3: There have been some reports on ZrTe₅, where SdH oscillations periodic in B (PRB 98, 165119, 2018), and log B (Sci. Adv. 4, eaau5096 (2018)) are discussed. Author should compare the cyclotron mass with these reports, and discuss the variation.

Authors: We thank the reviewer for this comment. We are aware of the work reported log B [Sci. Adv. 4, eaau5096 (2018)], but unaware of the polycrystal work claimed periodic-in-B relation [PRB 98, 165119 (2018)]. This work on polycrystalline ZrTe₅ reported basic physical properties, including preliminary clues of anomalous oscillations, but no conclusive experimental data, mass enhancement analyses, and flat band are reported partly due to polycrystals. We added this interesting work to our reference.

- A. The mechanism adopted by Sci. Adv. 4, eaau5096 (2018) is based on defect-induced Coulomb traps, where the bound states with log B relation scatter the Fermi level's mobile carrier sequentially. That mechanism is incompatible with the fermionic description of SdH oscillation based on the Lifshitz–Kosevich (L-K) formula. However, our results unambiguously show that the data strictly follow the L-K formula, which means that oscillations observed in our work are indeed SdH oscillations. Therefore, our work is different from [Sci. Adv. 4, eaau5096 (2018)], where no such effective mass fitting is found.
- B. On the other hand, the work on polycrystals referred by the Reviewer#3 already detected several wiggles on magnetoresistance, which is roughly linear-in-B due to the sample being highly oriented, although polycrystalline in nature. As shown in Fig. 4d of [PRB 98, 165119 (2018)], the effective mass around 9 T is 0.2 m_e , while our result is around 0.05 m_e . The polycrystal nature can explain this discrepancy. **One note is that this interesting work on polycrystal ZrTe₅ does not report the symmetry breaking, systematic enhancement of effective mass and the topological flat band, clearly distinguishing from our work.**

Reviewer #3: How does the plausibility of the topological flat band in the material makes it a 3D counter part of graphene which has Dirac cone shape band at Fermi level?

Authors: We thank the reviewer for this comment. Our paper claims the following logic: *1.* The pristine ZrTe₅ without polar distortion is a 3D counterpart of 2D graphene, although the pseudospin structure is different from the real spin nature in the Dirac models, but dispersions are all Dirac type; *2.* The flat band in graphene is achieved by moiré engineering; our work reports another route to realize a topological flat band in 3D Dirac material by polarity-assisted Rashba splitting under a magnetic field. The two parallel lines mentioned above drive us to conclude such a claim.

Summary of changes (All changes in main text and SI are highlighted in red):

1. We updated the related discussions on the mechanism of the observed anomalous quantum oscillations on page 9.
2. We added a detailed panel for crystal information, as shown in Fig. S1 in this updated version of SI.
3. We added the detailed analysis of Drude fitting as Fig. S2 in this updated version of SI.
4. The Standard SdH Landau fan diagram is plotted and added as Fig.S7 in the updated SI.
5. We updated the related discussions on A2 term by referring to the CVT results and SI on page 5.
6. We updated the method section in the main text to include more details on the caution of preparing the transport device on page 11.
7. References are updated.

Reviewers' Comments:

Reviewer #1:

Remarks to the Author:

The authors have responded to all questions raised by the referees. The materials added to the main text and the SI give a better understanding of the paper.

The authors added an essential reference [41] to answer the question about constant Fermi energy and improve the main text. In reference [41], it was shown that unusual SdH oscillations exist both in systems with constant Fermi energy and in systems with constant charge densities. The addition of this reference [41] makes the reader understand that the authors' conclusions are qualitatively correct in any case. The authors also modify and soften the main text about this issue.

However, in the response, the authors explain that they adopted the fixed Fermi energy approximation because the calculations of SdH for both scenarios look similar in [41]. Interestingly, the authors of [41] consider their calculations to be "quite different." Qualitatively, both scenarios indeed provide similar results, but numerically, the calculated SdH differ in amplitudes and peak positions. Are the SdH calculations similar for both scenarios in a flat-band case? This issue could be a good point for further theoretical investigations.

The paper represents a valuable experimental investigation with a qualitatively correct interpretation of the results. It will initiate further experimental and theoretical investigations.

Reviewer #2:

This is my second review report for the manuscript by D. Xing et al. I had pointed out two important issues in the first report:

1. The fundamental physics of using $\sin^2\{\theta\}$ in the second term while fitting the data. I understand the authors explained the fundamentals of torque magnetometry and what this technique can do, but I do not see any justification for using $\sin^2\{\theta\}$.
2. The major issue is the LK fit. I do not think this is the correct way to use the LK fit. Especially, the oscillations at lower field are not resolved. The authors pointed out that the fitting is good for high field points (B7 and B8) in S75, but whenever I looked at the raw data in the supplemental material [Fig. S9 (c)], all the quantum oscillation curves overlapped except one at 30 K. This means the temperature dependent data is not well-resolved for using the LK fit.

I recommend that the authors fix these issues before supporting publication.

Reviewer #3:

Remarks to the Author:

I am quite satisfied with the author's responses to my queries and they have made appropriate changes in the revised the manuscript. This is one of few manuscripts on ZrTe₅ which are on clean samples, and deserve publication. This study will further help in elucidating the exact behavior of this already enigmatic ZrTe₅ compound. I recommend the article for publication in Nature Communications.

Authors' reply to Reviewers' comments

Reviewer #1 (Remarks to the authors): The authors have responded to all questions raised by the referees. The materials added to the main text and the SI give a better understanding of the paper. The paper represents a valuable experimental investigation with a qualitatively correct interpretation of the results. It will initiate further experimental and theoretical investigations.

Authors: We thank the Reviewer #1 for the efforts/suggestions on this manuscript. We emphasize experimental data in our manuscript, on which the interpretation is formed. We agree with the Reviewer #1 that our understanding is qualitative, and the detailed band evolution (flattening) needs further spectroscopic and theoretical efforts.

Reviewer #1: However, in the response, the authors explain that they adopted the fixed Fermi energy approximation because the calculations of SdH for both scenarios look similar in [41]. Interestingly, the authors of [41] consider their calculations to be "quite different." Qualitatively, both scenarios indeed provide similar results, but numerically, the calculated SdH differ in amplitudes and peak positions. Are the SdH calculations similar for both scenarios in a flat-band case? This issue could be a good point for further theoretical investigations.

Authors: We thank the Reviewer #1 for this related comment. For '*but numerically, the calculated SdH differ in amplitudes and peak positions*', in 2D case, the authors in Ref. [41] pointed out that the difference originates from the gapless Hamiltonian (see Page 3 of Ref. [41]). However, in 3D case, the difference is not obvious, they claimed that the periodicity is similar (although peak positions are shifted) in both constraints of fixed energy and carrier (contrasted with doubled frequency for fixed carrier in 2D, Page. 3). However, the amplitude is quite different, on which we do not have a clear physical picture except an analytical expression.

The magnetic field-induced flat band here is interpreted within the Dirac model with adjustable effective mass, so the SdH calculations should be consistent with those in Ref. [41]. We also think the detailed band evolution with constraints from our experiment data should be a good point for further theoretical investigations.

Reviewer #2: The fundamental physics of using $\sin^2\{\theta\}$ in the second term while fitting the data. I understand the authors explained the fundamentals of torque magnetometry and what this technique can do, but I do not see any justification for using $\sin^2\{\theta\}$.

Authors: We thank the Reviewer #2 for this question about $\sin^2\{\theta\}$ term. For ZrTe₅, the undistorted crystal structure is *Cmcm* (No. 63) with a D_{2h} point group, which possesses three orthogonal C_2 rotation axes. Under the constraint of D_{2h} , the magnetic susceptibility tensor should be:

$$\chi_{D_{2h}} = \begin{pmatrix} \chi_{xx} & 0 & 0 \\ 0 & \chi_{yy} & 0 \\ 0 & 0 & \chi_{zz} \end{pmatrix}$$

However, this $\chi_{D_{2h}}$ cannot reconcile with the extra A2 term ($\sim \sin^2 \theta$), which means that the off-diagonal elements χ_{ij} must be nonzero. As we know, all the three point groups (D_2 , C_{2v} , D_{2h}) in orthorhombic structure make the off-diagonal χ_{ij} vanishing. We then have to go to structures with lower symmetry. As outlined in SI section I, we derived the torque expressions in monoclinic/triclinic polar structures with/without a single C_2 axis:

$$\vec{\tau}_{monoclinic} = \frac{1}{2} \mu_0 V H^2 \begin{pmatrix} \sin 2\theta \cdot (\chi_{yy} - \chi_{zz}) \\ \sin 2\theta \cdot (-\chi_{xy}) \\ \sin^2 \theta \cdot (2\chi_{xy}) \end{pmatrix}$$

$$\vec{\tau}_{triclinic} = \frac{1}{2} \mu_0 V H^2 \begin{pmatrix} (\chi_{yy} - \chi_{zz}) \cdot \sin 2\theta + 2\chi_{yz} \cdot \cos 2\theta \\ -\chi_{xy} \cdot \sin 2\theta - 2\chi_{xz} \cdot \cos^2 \theta \\ \chi_{xz} \cdot \sin 2\theta + 2\chi_{xy} \cdot \sin^2 \theta \end{pmatrix}$$

We can clearly see that the $\sim \sin^2 \theta$ (A2) term is solely related to the off-diagonal element χ_{xy} , which is absent in D_{2h} point group. However, as we discussed in the SI section I, the difficulty is that the monoclinic and triclinic structures adopt a non-orthogonal coordinate system, so we cannot tell which specific structure ZrTe₅ adopts in low temperature by torque measurement. We can only conclude that the low-temperature structure of ZrTe₅ is not orthorhombic and must have an off-diagonal element χ_{xy} .

Reviewer #2: The major issue is the LK fit. I do not think this is the correct way to use the LK fit. Especially, the oscillations at lower field are not resolved. The authors pointed out that the fitting is good for high field points (B7 and B8) in S75, but whenever I looked at the raw data in the supplemental material [Fig. S9 (c)], all the quantum oscillation curves overlapped except one at 30 K. This means the temperature dependent data is not wellresolved for using the LK fit.

Authors: We thank the Reviewer #2 for suggestions on L-K fitting. The main point is the temperature dependence of low-/high-field oscillations is not clear. We agree with the Reviewer #2 that the temperature dependence of B1, B2, B8, B9 should be clearly shown.

1. For low-field oscillations, the temperature dependence of oscillations is clearly resolved in S75 but not in S74. The reason is that the low-field background of S74 (Fig. 3c in the main text) differs from that of S75 (Fig. 3a), which makes extracting low-field oscillations of S74 difficult by exponential fitting. The problem can be solved by narrowing the fitting window. In Fig. R1, the low-field oscillations (B1, B2, B3) of S74 are now clearly resolved and show systematic temperature dependence.
2. For high-field oscillations, we measured the magnetoresistance of S74 and S75 again in the high-field facility center. As shown in Fig. R2, we now have new curves above 30 K. We also updated the L-K fit with the fitted values of effective mass.

The effective mass obtained from the new fittings is almost identical to the previous data. Based on newly fitted data, mass enhancement ~ 133 times at 25 T for S74, and ~ 90 times for S75 (see Fig. R1 and R2). For consistency of B1 B2 B3 of Fig. 4b in the main text, we update the caption of Fig. 4 that the precise L-K fitting can be found in SI (Fig. S10).

Fig. R1 Temperature dependence of low-field oscillations in S74. **a**, Temperature-dependent $\Delta\rho_{xx}$ (smoothed) of S74 below 0.5 T. **b-d**, The L-K formula fittings for B1, B2 and B3 data.

Fig. R2 Additional data of temperature-dependent $\Delta\rho_{xx}$. **a**, Temperature-dependent $\Delta\rho_{xx}$ of S75. **b**, The L-K formula fittings for data from **a**. **c** & **d**, Temperature-dependent $\Delta\rho_{xx}$ of samples S74 and S75 measured in high fields. **e** & **f**, The L-K formula fittings for high-field oscillations

Reviewer #3 (Remarks to the authors): I am quite satisfied with the author's responses to my queries and they have made appropriate changes in the revised the manuscript. This is one of few manuscripts on ZrTe5 which are on clean samples, and deserve publication. This study will further help in elucidating the exact behavior of this already enigmatic ZrTe5 compound. I recommend the article for publication in Nature Communications.

Authors: We thank the Reviewer #3 for the recommendation for publication.

Summary of changes (All changes in main text and SI are highlighted in red):

1. Fig. 4 in the main text is updated by replacing the previous L-K fitting on B8 and B9. The caption of Fig. 4 has also been updated.
2. Fig. S9 is updated by adding newly measured high-field data on S74 and S75, and L-K fittings are also added to Fig. S9.
3. Fig. S10 is added to SI to exhibit the resolved low-field oscillations in S74.
4. Fig. 5 in the main text is also updated with the newly fitted values of effective mass.

Reviewers' Comments:

Reviewer #1:

Remarks to the Author:

I think that the paper has enough new and important experimental data for publication. I am satisfied with the authors answer on my remark. I agree that the field-induced flat-band model is an interesting and reasonable explanation.

Reviewer #2:

Remarks to the Author:

Only confidential comments to the Editor were provided.

Authors' reply to Reviewers' comments

Reviewer #1 (Remarks to the authors): I think that the paper has enough new and important experimental data for publication. I am satisfied with the authors answer on my remark. I agree that the field-induced flat-band model is an interesting and reasonable explanation.

Authors: We thank the Reviewer #1 for the recommendation for publication.